# CoMIX: A Multi-agent Reinforcement Learning Training Architecture for Efficient Decentralized Coordination and Independent Decision-Making

**Giovanni Minelli**                                          *giovanni.minelli2@studio.unibo.it*
*University of Bologna*

**Mirco Musolesi**                                                    *m.musolesi@ucl.ac.uk*
*University College London*
*University of Bologna*

**Reviewed on OpenReview:** *https://openreview.net/forum?id=JoU9khOwwr*

## Abstract

Robust coordination skills enable agents to operate cohesively in shared environments, to-gether towards a common goal and, ideally, individually without hindering each other's progress. To this end, this paper presents Coordinated QMIX (CoMIX), a novel train-ing framework for decentralized agents that enables emergent coordination through flexible policies, allowing at the same time independent decision-making at individual level. CoMIX models selfish and collaborative behavior as incremental steps in each agent's decision pro-cess. This allows agents to dynamically adapt their behavior to different situations balancing independence and collaboration. Experiments using a variety of simulation environments demonstrate that CoMIX outperforms baselines on collaborative tasks. The results validate our incremental approach as effective technique for improving coordination in multi-agent systems.

## 1 Introduction

Multi-agent systems pose significant challenges due to the complexity of managing interactions (Long Ma et al., 2015). Whether agents operate individually or collaboratively, coordination emerges as a pivotal aspect within such systems. Moreover, as the number of agents increases, the task of ensuring coordination becomes harder and harder. Effective coordination is also critical for a number of practical application domains, namely autonomous driving (Prabuchandran et al., 2014; Sadigh et al., 2018; Falsone et al., 2020) and robotics (Yan et al., 2013; Yu et al., 2023), especially in systems involving mobile robots (Chen & Luh, 1994; Sousa et al., 2021) and drone fleets (Alfeo et al., 2018; Qin & Pournaras, 2023).

Previous research has focused primarily on information sharing strategies to facilitate coordination among multiple entities (Groen et al., 2007; Pesce & Montana, 2020; Kim et al., 2021). Indeed, agents often find themselves having to make decisions based on an incomplete understanding of the world in which they operate, for instance, due to limited perception ability or scarce computational resources - these are typical examples of *bounded rationality* (Simon, 1996). In this context, effective communication plays a key role. In fact, coordination emerging from indiscriminate communication may not always be the best solution(Cason et al., 2012): an overabundance of shared information can overwhelm agents and slow down decision-making, negatively affecting coordination among the agents.

In addition, there are cases where coordination itself can hinder optimal task execution. For example, popular solutions, such as IC3Net (Singh et al., 2019), BicNet (Peng et al., 2017), and ATOC (Jiang & Lu, 2018), do not explicitly grant agents the autonomy to operate outside the given cooperative scheme by limiting the exploration of the entire action space and the exploitation of individual capabilities. Allowing

agents to occasionally pursue local strategies could be beneficial to achieve the overall shared collective goal, as agents can help discover better optimized solutions for the team as a whole. For example, in a multi-agent navigation task, a robot periodically exploring new potential paths on its own could provide useful information to others. This raises an important question that we aim to address in this study: *can agents acting in a shared environment, through communication and without collaboration constraints, learn to balance coordinated and individual behaviors in order to successfully achieve a given goal?*

In order to address this problem, in this paper, we present the design, implementation and evaluation of *Coordinated QMIX (CoMIX)*[1], a novel framework for decentralized multi-agent reinforcement learning (MARL) combining group and individual decision-making. The conventional approach to addressing the coordination problem in MARL systems is to prioritize group choices over individual ones. Although a solution that gives more importance to group dynamics can be more effective when cooperation is critical for success, it might be detrimental when agents could benefit from acting independently to achieve a given goal, or in complex environments with a vast space of possible interactions. For instance, in crowded or sparsely rewarded settings, enforcing top-down coordination on all agents may hinder convergence to an optimal joint policy (Aotani et al., 2018). In contrast, allowing potential independent local choices, rather than mandating group coordination, can lead to emergent collaboration arising naturally where needed, with benefits for both the individual and the group as a whole (Kar et al., 2013; Hughes et al., 2018; Yang et al., 2020; Yu et al., 2022).

CoMIX embodies this principle by allowing agents to choose whether to co-operate with others or to act independently, first communicating their own action intentions derived from previously acquired and current information alone (e.g., sensory data). These are then combined with the intentions of other agents in order to reduce uncertainty and achieve coordination.

By considering a variety of challenging MARL environments, we demonstrate CoMIX's ability to efficiently choose between collaboration and independent action, and, consequently, to coordinate with a large number of agents effectively. Moreover, CoMIX is based on storing a weighted record of neighbors' previous action intentions for coordination, which makes the method interpretable in terms of agent choices and resilient to communication disruptions. Overall, our findings highlight the effectiveness of CoMIX in facilitating coordination among agents, by enabling them to selectively coordinate in pursuit of a shared goal, allowing at the same time independent decision-making when beneficial.

## 2 Related Work

In this section, we discuss the relevant related work with a focus on mechanisms for group coordination and efficient information sharing in multi-agent systems.

**Coordination.** Multi-Agent Reinforcement Learning (MARL) frameworks are designed to overcome the added complexity resulting from the presence of multiple interacting entities in an environment (Tuyls & Weiss, 2012; Zhang et al., 2021). In particular, we are interested on those that adopt information sharing mechanisms (Gronauer & Diepold, 2021; Hernandez-Leal et al., 2019) to achieve flexible coordination without introducing central entities that could create failure points or bottlenecks. CommNet (Sukhbaatar et al., 2016) demonstrates remarkable performance by simply collecting messages and averaging them, then using the additional information in each agent's policy. IC3Net (Singh et al., 2019) adds a mechanism for instructing agents to learn when to communicate, through the adoption of a gating mechanism to allow access to a communication channel. However, poor utilization of the communication channel limit the ability to coordinate and may not lead to effective policies when many agents participate in communication. BicNet (Peng et al., 2017) uses a more complex mechanism to leverage incoming information by adopting a bidirectional module, and, like others approaches (Jiang & Lu, 2018; Das et al., 2019), is characterized by an architecture based on recurrent modules to maintain consistency in outputs generated in subsequent steps. However, these solutions are designed only for cooperative tasks (Mao et al., 2017; Foerster et al., 2018; Singh et al., 2019; Zhang et al., 2020; Li et al., 2022) and, therefore, lead to policies not considering the independence of agents or allowing them to exploit possibly better solutions by diverging from team

---

[1]The code is available at the following URL: https://github.com/johnMinelli/CoordinatedQMIX

executions. Our approach builds on this body of work, with a specific focus on flexible coordination and communication efficiency.

**Information Sharing.** Communication can be implemented as a direct message sent through a private channel (Niu et al., 2021), or as a broadcast transmission (Sukhbaatar et al., 2016; Lin et al., 2021). However, the latter solution may prevent agents from distinguishing information that is valuable for them from irrelevant or even potentially misleading one in case of different action dynamics or goals. In general, the information received can be considered valuable if it enhances an agent's understanding and coordination by providing relevant updates on the environment state, goals, intentions, and other factors that aid decision-making and coordination. ATOC (Jiang & Lu, 2018) proposes a communication model with a gate mechanism for communication efficiency. Instead, (Das et al., 2019; Li et al., 2022; Kim et al., 2021) act on the listener's side, adopting different attention mechanisms to filter out irrelevant messages. However, they generally consider each message in isolation rather than in the context of all the others in the communication channel, missing opportunities to coordinate responses among multiple agents as it is often essential for effective teamwork. Some approaches rely on the formation of groups of agents and focus on inter- and intra-communication to limit computational efforts and improve coordination performance (Jiang & Lu, 2018; Liu et al., 2020; 2021; Niu et al., 2021), while others reduce the number of messages sent by learning in which situations communication is really necessary and/or when the available information is redundant and therefore communication can be avoided over time (Ding et al., 2020; Liu et al., 2020; Kim et al., 2021; Kalinowska et al., 2022). In contrast, CoMIX aims at creating simplified communication channels directly based on environment observations, not requiring agents to make assumptions about the senders or receivers, thus making them potentially deployable even with unseen agents.

**Information Representation.** The format of a message is a crucial aspect of communication, as it must effectively encode information in order to reduce uncertainty and facilitate coordination with the recipient. The information conveyed through the message may encompass various aspects of communication and decision-making, ranging from agent intentions, such as requests, questions, and plans, to contextual information and conversational messages. Some existing work relies on the transmission of internal state vectors (Sukhbaatar et al., 2016; Peng et al., 2017; Singh et al., 2019; Wang & Sartoretti, 2022), which are typically used by agents to encode their own history, but the reuse of such individual state representations for communication may not be suitable. In (Das et al., 2019; Li & Zhang, 2024), targeted messages are used to convey information to individual agents. Simpler communication approaches (Liu et al., 2020; Kim et al., 2021) include current or time-delayed observations – plain or encoded – as content for the communication message. This allows for more flexible interaction dynamics since the information is transmitted without any additional processing or filtering. CoMIX belongs to this class of solutions, since it relies on sharing observations paired with the agent's action decision, in order to guide coordination among agents.

## 3 Background

In this section, we introduce the notation and concepts at the basis of our approach, which will then be presented in detail in the following section.

**Deep Reinforcement Learning.** The problem is modeled as a decentralized partially observable Markov decision process (Dec-POMDP) (Oliehoek & Amato, 2016), which is defined by the tuple $(K, S, \mathcal{A}, P, \mathcal{R}, \gamma)$, where $K$ is the set of $n$ interacting agents, $S$ is the set of observable states $s$, $\mathcal{A}$ is the joint action space consisting of individual action spaces $A_i$ of actions $a$, for each agent $i \in K$, $P : S \times A \to P(S)$ is the transition probability function that describes the chance of a state transition, $\mathcal{R}$ is the joint set of reward functions $R_i$ associated with each agent $i \in K$, and $\gamma \in [0, 1)$ is the discount factor. At each time step, each agent selects an action based on its individual policy $\pi_i : S \to P(A_i)$. The system evolves from joint state $\mathbf{s} = \{s_1, \ldots, s_n\}$ to the next state $\mathbf{s}'$ under the joint action $\mathbf{a} = \{a_1, \ldots, a_n\}$ with respect to the transition probability function $P$, while each agent receives an immediate reward $\mathbf{r} = \{r_1, \ldots, r_n\}$ as feedback following a state transition. The objective is to construct an independent policy $\pi_i$ for each agent $i \in K$ with the goal of maximizing the expected discounted return. The optimal policy $\pi^*(s)$ is obtained indirectly during the iterative process by considering a state-action function $Q_i : S \times A_i \to R_i$, associated with each agent, which

estimates the expected return of taking an action $a$ in a state $s$. The learning algorithm is based on a Deep Q-Network (DQN)(Mnih et al., 2013): the Q-function is parametrized by a neural network presenting recurrent neural network (RNN) modules to keep memory of sequences of choices. The Dec-POMDP setting is then extended with a broadcast communication channel allowing agents to exchange messages to overcome partial observability effects. The individual policies are learned as state-action functions $Q_i(s_i, h_i, \mathbf{m}, a_i)$ evaluating the possible actions $a_i$ in the current local state $s_i$ with the hidden state $h_i$, and the exchanged messages $\mathbf{m} = \{m_1, \ldots, m_n\}$, where $m_i$ is sent by agent $i \in K$.

**Centralized Training with Decentralized Execution.** Centralized training with decentralized execution (CTDE) with Q-value optimization is a popular approach for solving MARL problems (Lowe et al., 2017; Foerster et al., 2018; Rashid et al., 2020). During training, a central mixer network is used to represent the value of the state of the entire system. We adopt QMIX (Rashid et al., 2020), which learns a non-linear mixing function to map individual Q-values of agents $\mathbf{q} = \{Q_1, \ldots, Q_n\}$ to the global Q-value, denoted as $Q_{TOT}$. Hypernetworks (Ha et al., 2016) are used to compute a set of mixing weights for each agent, ensuring that a global argument of the maxima (argmax) operation produces the same result as individual argmax operations. This is achieved through a monotonicity constraint between $Q_{TOT}$ and individual value $Q_i$:

$$\frac{\partial Q^{TOT}}{\partial Q_i} \geq 0, \forall i \in [i, n] \tag{1}$$

## 4 The CoMIX Training Architecture

In the following we will describe CoMIX, a framework that allows agents to adopt both individual and group strategic behavior by selectively communicating via a shared channel, letting coordination arise from continuous interaction and without making prior assumptions about other agents. The overall architecture of CoMIX is depicted in Figure 1.

### 4.1 Architecture

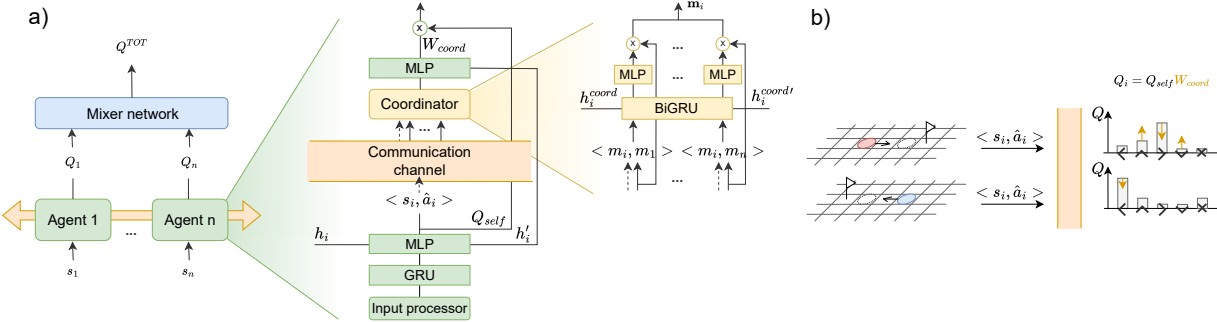

Figure 1: CoMIX training architecture. (a) The system is illustrated by breaking down its components with the information flow from left to right. Each agent observes a partial state of the world, processes it, and determines the next action to be taken. After transmitting this information $< s_i, \hat{a}_i >$ through the communication channel, they receive and filter the $\mathbf{m}$ messages from other agents using a coordination module. The filtered messages $\bar{\mathbf{m}}_i$ are then used to compute weights to rescale the original state-action pairs and select the best action. Decisions are made locally. (b) A conceptual demonstration of the interpretability of the method showing how two agents, sharing the same portion of the map and with opposite goal directions, change their initial action intentions based on information received from other agents.

#### 4.1.1 Action Policy

The action policy in our approach involves predicting action values as element-wise product of two terms: $Q_i = Q_{self} \odot W_{coord}$. This computation occurs within the Q-network through a two-step process. In the

first step, $Q_{self}$ is obtained by considering only the internal state of the agent $h_i$ and its current observation $s_i$. This step does not involve communication with other agents. After communication occurs, the second term $W_{coord}$ is computed within the Q-network to weigh the estimated value for each action. Specifically, $W_{coord}(h_i', \bar{\mathbf{m}}_i, a_i)$ takes into account the updated hidden state $h_i'$ and the messages filtered by the Coordinator module for agent $i$. We will discuss these aspects in detail in the next section. The messages are averaged in order to form a single vector concatenated with the hidden state and processed through a two layer fully connected network to obtain a bounded weight value. It is worth noting that the raw observation is first passed through an input processing module, which extracts features and reduces the state space. Following the approach in Lin et al. (2021), shared weights are used among all agents to allow them to reason about data from the same distribution. Finally, each output $Q_i$ (for each agent $i$) of the Q-network is defined as follows:

$$Q_i(s_i, h_i, \bar{\mathbf{m}}i, a_i) = Q_{self}(s_i, h_i, a_i) \odot W_{coord}(h_i', \bar{\mathbf{m}}_i, a_i) \tag{2}$$

### 4.1.2 Coordinator

The Coordinator module is responsible for determining the relevance of other agents' communications in relation to the agent's intentions by generating a coordination mask used to filter out incoming messages. A message is defined as the communicated intention of an agent to take a certain action in its current state to achieve an objective, $\hat{a}_i = \arg\max_a Q_{self}(s_i, h_i, a)$, and it is represented by the tuple $m_i = <s_i, \hat{a}_i>$. Consequently, we define $\mathbf{m} = \{m_1, \dots, m_n\}$ as the set of incoming messages sent by $n$ agents. As previously proposed by (Jiang & Lu, 2018), the Coordinator module is implemented using a BiGRU layer (Wang et al., 2017) and an additional Linear layer on top. A two-way softmax operation then extracts the probability of accepting or rejecting each message as shown in Eq.4:

$$\mathbf{z}_i = \{<m_i, m_1>, \dots, <m_i, m_n>\}^{-<m_i, m_i>} \tag{3}$$

$$\mathbf{c}_i = Coord(\mathbf{z}_i) \tag{4}$$

where $\mathbf{c}_i = \{c_{i,1}, \dots, c_{i,n}\}^{-c_{i,i}}$ and $c_{i,j} = \begin{cases} 1 & \text{if agent } i \text{ coordinate with agent } j, \\ 0 & \text{otherwise} \end{cases}$

We then obtain $\bar{\mathbf{m}}_i$, the filtered set of messages for agent $i$, by applying the Hadamard product between incoming messages and the coordination mask $\mathbf{c}_i$ computed by the Coordinator module:

$$\bar{\mathbf{m}}_i = \mathbf{m} \odot \mathbf{c}_i \tag{5}$$

### 4.2 Training

### 4.2.1 Centralized Temporal Difference Supervision

The agents' Q-policy networks are trained end-to-end by adopting a CTDE framework that estimates the state-action value of the entire system. We adopt the QMIX framework (Rashid et al., 2020), which despite the monotonicity constraint limiting its representation capacity, has been shown to achieve state-of-the-art performance on various MARL tasks. Additionally, its computational simplicity enables it to effectively manage large joint action spaces. We consider $\theta^Q$ that parameterizes both the components of individual agents responsible for estimating the Q-value and the central mixer network. Following the popular implementation of Rashid et al. (2020), we adopt the time difference error as the update rule:

$$L_Q(\theta^Q) = |y^{TOT} - Q^{TOT}(\mathbf{s}, \mathbf{q}; \theta^Q)|, \tag{6}$$

where $y^{TOT} = \mathbf{r} + \gamma \max_{\mathbf{q}'} Q^{TOT}(\mathbf{s}', \mathbf{q}'; \theta^{Q'})$ and $\theta^Q$, $\theta^{Q'}$ are respectively the parameters of the online policy network and target policy network, periodically copied from $\theta^Q$ as in standard DDQN (Van Hasselt et al., 2016). Employing a centralized function that accounts for all agent actions can lead to a decrease of the policy gradient estimate variance compared to using individual Q-networks per agent, as separate Q-networks may inaccurately assess the local Q-function. Then, after training, the central training function is not used anymore, and policy execution is fully decentralized.

### 4.2.2 Contrastive Optimization

The parameters of the Coordinator module are updated using a contrastive optimization scheme. The underlying idea is that providing the individual policy with helpful information could enable it to more accurately predict the expected future rewards associated with each action. Assuming to be able to achieve an approximation of the optimal Q-function, we aim to train an efficient coordinator. This coordinator acts as a filter over the communication channel, selecting the subset of messages used to calculate the state-action value estimate with maximum cumulative expected reward. We update the Coordinator's parameters $\theta^C$ by extracting the loss signal as the clipped difference between the maximum value of the state-action pairs obtained with filtered messages $\bar{\mathbf{m}}_i$ using the predicted coordination mask $\mathbf{c}_i$ and the estimated value obtained with an alternative subset of messages $\tilde{\mathbf{m}}_i = \mathbf{m} \odot \tilde{\mathbf{c}}_i$, where $\tilde{\mathbf{c}}_i = (1 - \text{Coord}(\mathbf{z}_i; \theta^C))$ from Eq. 4 and Eq. 5. Notably, we propagate the error in the probabilities of message selection $\mathbf{c}_i$ and keep the weights of the Q-networks fixed by introducing a stop-gradient operator `stop`:

$$L_C(\theta^C) = \sum_{i=1}^{n} \texttt{stop}(w_i \Delta Q_i)\mathbf{c}_i = \sum_{i=1}^{n} \texttt{stop}(w_i \max(0, \max_{a_i} Q_i(s_i, h_i, \tilde{\mathbf{m}}i, a_i) - \max_{a_i} Q_i(s_i, h_i, \bar{\mathbf{m}}_i, a_i)))\mathbf{c}_i \quad (7)$$

The additional weighting term $w_i$ is obtained by means of the QMIX Mixer network, which, by design, projects the individual Q-value estimates from each agent into a shared representation space in its first layer. We leverage the monotonicity constraints of the Mixer to compute $w_i$ as the per-agent averaged projection weights, which scale their estimated Q-values based on the current joint state.

### 4.2.3 Training Details

We keep the size of the hidden layers fixed at 128 for all submodules of the Q-networks. The two hidden layers of the Mixer network are configured with 32 and 16 nodes, respectively. We use ReLU nonlinearity and apply layer normalization after both recurrent modules – in the Q policy network and in the Coordinator. This helps address scalability issues caused by vanishing gradients when dealing with many agents. To optimize the weights of the Q and Coordinator networks, we use RMSprop. The learning rate is set to $1e^{-4}$ for the Q-network and $5e^{-5}$ for the Coordinator network. We also use a weight decay value of $1e^{-5}$ to prevent catastrophic forgetting and overfitting. To stabilize Q-learning, we update the weights of the target networks every 100 episodes. The batch size and number of recurrent steps during training are chosen based on the complexity of the task. Additional details about networks structure, hyperparameters optimization, the simulation environments, and the algorithm at the basis of the CoMIX training loop can be found in the Supplementary Material.

## 5 Experiments

We evaluate CoMIX in three distinct environments, which allow us to assess different dimensions of the problem, including scalability and robustness to potential disruptions.

### 5.1 Environments

In this subsection, we introduce the environments used in our experimental evaluation, providing an overview of their main characteristics and discussing the criteria for their selection. For a full description of the experimental settings of the environments, please refer to the Supplementary Material.

### 5.1.1 Switch

Switch is a gridworld environment in which 4 agents spawn in corners and must reach target positions on the opposite side of the map. It consists of a rectangular grid with a long narrow corridor in the center which can only be crossed by a single agent at a time. The vertical position of the corridor is randomized for each episode to increase unpredictability. The reward is given only upon reaching the destination, and its value is determined by how quickly the task is completed. This environment can specifically be used to evaluate the agents' abilities to pursue a long term reward and balance aggressive and coordinated strategies since

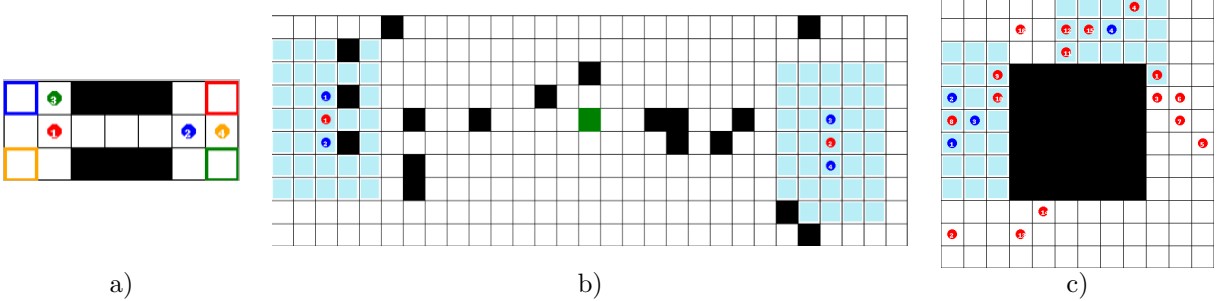

Figure 2: Environments used for experimental evaluation. From left to right (a) Switch; (b) Cooperative Load Transportation, 2 loads (red) and 4 agents (blue) with an equal distance of 15 steps to the final position (green), and 10% of randomly selected cells being obstacles (black); (c) Predator-Prey, 4 agents (blue), 16 prey (red) in a 12x12 map. The spatial observability of agents is shown in light blue.

taking the initiative to traverse the corridor earlier leads to better rewards, but risks potential gridlock if agents collide. The agents can only observe their current position and target distance, but not other agents, so they must communicate to coordinate. Therefore, in order to complete the task, it is necessary to balance individual and collaborative behavior. In addition, we would also like to note that partial observability forces agents to communicate and adapt their policies based on the actions of others.

### 5.1.2 Cooperative Load Transportation

This environment involves pairs of agents synchronizing to transport a load to a goal location while avoiding obstacles. It requires close coordination to move in the same direction and effective communication to prevent interference from other agent pairs. Agents observe their absolute coordinates, distance to the goal, and proximity to obstacles. Small rewards are given for moving the load closer to the goal and a large reward upon completion. We select this environment since it requires very fine-grained coordination and the capability of screening out "distracting" signals from other agents.

### 5.1.3 Predator-Prey

This is a classic MARL environment (Gupta et al., 2017) adapted to encourage interactions between agents. The environment is a grid world in which agents, the predators, chase randomly moving entities, the prey, with the aim of capturing them. The predators' visibility is limited to a range of two units in each direction from their positions, and they move at the same speed as the prey. The agents are sparsely rewarded. The main challenge lies in collaborating as a group to surround the prey on all four axes: in this case an agent receives a substantial reward. If the agents act independently, they can receive smaller rewards by attempting to occupy the same cell as the prey. This task is relatively straightforward and serves to distract the agents, as the ultimate measure of success is the tally of captured prey. Therefore, the agents should prioritize maximizing long-term cumulative rewards through coordinated hunting and accomplishing the challenging group task, rather than focusing on easily attainable short-term individual rewards.

### 5.2 Baselines

For the evaluation, as comparators, we select a straightforward "vanilla" implementation of PPO in a multi agent setting, *IPPO* (Schroeder de Witt et al., 2020), and two representative MARL methods that implement attentional communication, namely *IC3Net* (Singh et al., 2019) and *ATOC* (Jiang & Lu, 2018). In particular, the choice of PPO, is motivated by its demonstrated ability to excel in certain scenarios characterized by specific forms of environmental nonstationarity, thus allowing IPPO to exhibit robustness when dealing with cooperative tasks that can be solved through sequences of independent actions selected locally. Compared with ours, IC3Net utilizes a gating mechanism on incoming communications, wherein it aggregates all additional information by computing the average into a single vector that is then used in the internal

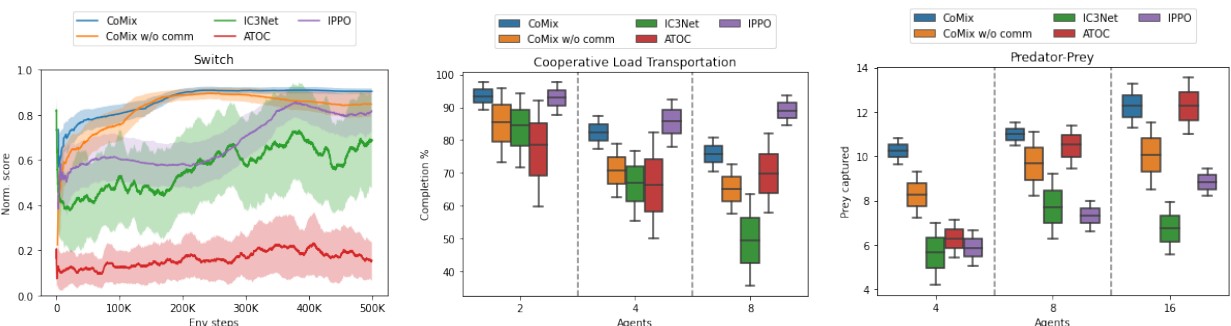

Figure 3: Training results in the three environments used for evaluation. We adopt the following performance metrics: in Switch, the sum of rewards obtained by all agents normalized to 1; in Cooperative Load Transportation, the distance of the load from the docking area as the percentage of task completion; in Predator-Prey, the number of prey captured.
For a fair comparison, we kept the number of prey constant at 16 by scaling the map size and the number of agents, respectively: 12x12 with 4 agents, 14x14 with 8 agents, and 16x16 with 16 agents. The standard deviation is relative to the mean of the results of 5 different seeds.

policy to generate two output heads: one for the actions' probabilities and the other for the individualized probability of communication. On the other hand, ATOC adopts a two-step policy: first reasoning about incoming partial observations, then forming semi-persistent communication groups by considering each other agent individually, and employing a BiLSTM (Graves & Schmidhuber, 2005) module to compute the final action.

## 5.3 Experimental Results

Figure 3 shows a performance comparison between CoMIX, the baselines and "CoMIX w/o comm", an ablated version without direct communication between agents whose architecture corresponds to the implementation of QMIX. Comparison with the latter and with IPPO, in which agents rely solely on local observations to estimate their value function, also highlights the importance of the communication channel. In the Switch environment, the agents do not require complex strategic actions. In this case, communication is mainly used to signal the presence of an agent. However, different agent behaviors can lead to very rapid task completion or longer episodes due to collisions and mutual blocking of agents, which increase the variance of reported results. In this case, CoMIX consistently enables all agents to learn behaviors that allow for faster task resolution. Instead, IC3Net, adopting a simpler approach to coordination, learns deterministic action sequences causing agents to behave with much variability. In contrast, ATOC's performance is affected by its inability to handle policies with conflicting goals, which is a consequence of weight sharing between agents. Finally, IPPO shows competitive performance but bounded by the partial observability of the agents. We further evaluate CoMIX considering the Cooperative Load Transportation and Predator-Prey tasks at incremental scales reporting the average performance and standard deviation for all methods. In Cooperative Load Transportation, CoMIX outperforms ATOC and IC3Net baselines, demonstrating good abilities in reaching the goal location avoiding obstacles along the way. Even with an increasing number of agents, we demonstrate that it is able to maintain good performance overall by filtering the communication channel from irrelevant messages coming from other agents. IPPO, on the other hand, shows clear superiority in resolution, being unaffected by external communication noise and taking advantage of the lack of marked randomness in the task objective. In the last environment, CoMIX learns how to coordinate both small and large numbers of agents by covering incremental portions of the map, whose size scales accordingly, while the number of preys remains constant. Interestingly, while ATOC does not perform well in coordinating few resources, it becomes competitive as the number of agents present in the map increases. Meanwhile, both IC3Net and IPPO underperform as agents act strictly locally. In general, we notice that CoMIX is able to perform consistently across experiments. Additionally, we examine the importance of the communication component in the proposed method with a direct comparison with its ablated version. Despite the Mixer network still

allows for information sharing at a global level, we notice how tasks like Load Transportation and Predator-Prey are affected by a decrease in overall performance. This highlight the importance of local choices in such environments and the potential advantages of incorporating external information into the decision-making process. We observe that collaborative scenarios are infrequently observed. For example, in Cooperative Load Transportation, avoiding an obstacle leads to numerous instances of movement disagreement, while in Predator-Prey, agents prioritize local actions instead of forming teams. We also test a scenario in which agents are initially deployed at distant locations in a map with no prey present. By observing the exploration abilities of the agents, we notice that agents involved in communication often converge into groups compared to those that are not.

Table 1: Effects of communication channel disruption on the performance metric adopted for each environment. The performance metrics are as follows: in Switch, the sum of rewards obtained by all agents normalized to 1; in Cooperative Load Transportation, the distance of the load from the docking area as the percentage of task completion; in Predator-Prey, the number of prey captured. We show the performance of our method in relation to the reduction in the amount of messages exchanged and the results after optimization process to handle missing or delayed messages. Depending on the environment, we are able to achieve better performance in terms of number of messages transmitted.

| Comm channel usage | Switch | | Transport (4 agents) | | Predator-Prey (4 agents) | |
|---|---|---|---|---|---|---|
| | baseline | +fine-tuning | baseline | +fine-tuning | baseline | +fine-tuning |
| 100% - Full comm | 0.905 | $\sim$ | 82.07 | $\sim$ | 10.85 | $\sim$ |
| 50% | 0.903 $\sim$ | 0.907 $\sim$ | 79.07 ↓ | 82.58 ↑ | 10.78 $\sim$ | 10.90 $\sim$ |
| 25% | 0.904 $\sim$ | 0.903 $\sim$ | 78.64 ↓ | 83.45 ↑ | 10.36 ↓ | 11.15 ↑ |
| 10% | 0.898 $\sim$ | 0.905 $\sim$ | 77.38 ↓ | 84.06 ↑ | 10.10 ↓ | 10.82 $\sim$ |
| 0% - No comm | 0.900 $\sim$ | 0.907 $\sim$ | 77.78 ↓ | 79.35 ↓ | 10.34 ↓ | 10.70 $\sim$ |

## 5.4 Analysis

### 5.4.1 Efficiency

An efficient algorithm should learn to avoid unnecessary communications, thus reducing the effects of information overload. Figure 4 shows how many messages are typically considered when agents are asked to coordinate with others. Average usage varies depending on the task at hand, and we emphasize that we achieve this behavior by fully learning communication without imposing external constraints. In line with the task rules, we observe significant reductions in the number of messages used in Switch and Cooperative Load Transportation. More interestingly, we also notice similar behavior for the Predator-Prey task: although all agents share the general goal of capturing prey, the policy restricts each agent to communicating with only a subset of other agents, enabling the formation of smaller, more efficient subgroups to coordinate actions rather than requiring communication across every actor. As a further evaluation, we added "noisy" agents to the environments, which send a sequence of randomly generated bits, thus increasing the noise in the communication channel in the form of irrelevant messages. We note that their introduction does not substantially affect the average utilization of the communication channel, as expected, since such agents do not contribute to the solution of the task.

### 5.4.2 Communication

Since packet drops and intermittent availability of communications are not uncommon in the real world, we studied the ability of CoMIX to operate with a disrupted communication channel, in which messages are not delivered at every step. Table 1 reports as baseline the results of average performing models trained in normal communication settings and evaluated reducing messages exchanges, up to complete drop of the communication channel. Sequences of messages are dropped randomly in the process, providing in their place "outdated" messages, i.e., the latest information transmitted by the agent. As expected, we notice a slight drop in performance that increases as communication decreases. On the other hand, fine-tuning the policy

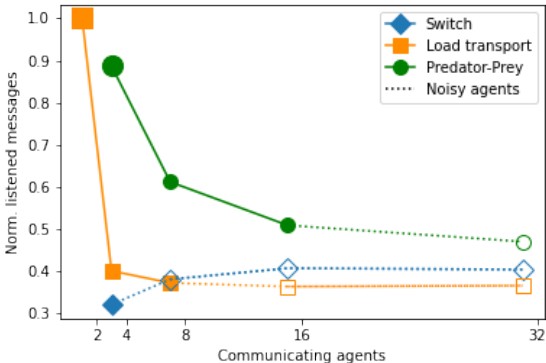

Figure 4: Average number of messages accepted by each agent's Coordinator normalized by the number of real agents acting in the environment. By introducing fictitious "noisy agents" transmitting random bits into the communication channel, we expect the value to remain unaltered.

weights according to this new setting, we reverse the trend and even achieve improvements over the initial performance, as evidenced in Table 1. To implement this procedure, we freeze the Coordinator weights, and randomly discard sequences of messages during training. Features of the messages communicated in their place (after two linear layers of processing) are then scaled proportionally to the delay times to reflect their temporal relevance. The process of fine-tuning the weights is performed until the performance matches the starting model under normal communication conditions, using a learning rate reduced by two order of magnitude. Again, Switch agents demonstrate to not rely particularly on the communication channel, while in the Predator-Prey and Cooperative Load Transportation environments the fine tuning allows improved results with less communication, reaching a reduction of -90% in the latter. We also highlight the comparison of agents trained with communication enabled, but then disabled at run-time (Table 1, last row), versus agents trained with no communication at all (Figure 3): message exchange enables agents to acquire skills, one can take advantage even in situations of no communication.

### 5.4.3 Flexibility

Compared with existing approaches that impose strict coordination, CoMIX offers greater flexibility and autonomy at the individual agent level, while still achieving effective coordination when needed. This flexibility is particularly advantageous for tasks where agents can operate independently for periods of time, but need to collaborate effectively when opportunities arise. In the Predator-Prey environment, we observe examples of this flexible behavior emerging. Agents would often spread out initially to explore the space independently. However, when prey is detected, the agents demonstrate the ability to dynamically coordinate their actions, surrounding and capturing them through cohesive collaboration. This flexibility allow the agents to effectively explore the large joint action space rather than being constrained to strict coordination protocols. This does not have a negative impact on CoMIX's coordination performance as demonstrated by the experimental results. In Predator-Prey, CoMIX significantly outperforms baselines like IC3Net and IPPO that either enforce rigid communication structures or lack such mechanisms (Figure 3). Notably, CoMIX adjusts its communication and coordination levels based on task demands. It employs sparser communication in the Switch environment, while utilizing higher levels in Load Transport and Predator-Prey (Figure 4). The flexibility of CoMIX is especially beneficial in sparse rewards settings. Without enforcing over-constrained collaboration, in settings like the one of Predator-Prey with only 4 agents, CoMIX enables faster convergence, compared to baselines. At the same time, by allowing for independent local choices, it sidesteps the issue of insufficient exploration of individual actions, as evidenced by the performance in the Switch environment. In general, the proposed method is suitable for all scenarios in which agents can effectively use a communication channel to coordinate their actions, because of its flexibility and robustness to noisy communication channels.

# 6 Conclusion

In this paper, we have introduced CoMIX, a new training architecture that effectively addresses coordination in multi-agent systems by separating selfish from collaborative behaviors. Through an extensive experimental evaluation by means of a variety of environments, we have shown the applicability of CoMIX to different tasks, and we have highlighted its ability to improve the performance of individual agents' policy in performing collaborative tasks in a decentralized fashion. In addition, we have demonstrated the robustness of our approach to communication disruptions and its effectiveness in filtering unnecessary information before it is transmitted: these are important characteristics for potential real-world deployments.

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
