# CoMIX: A Multi-agent Reinforcement Learning Training Architecture for Efficient Decentralized Coordination and Independent Decision-Making
# Supplementary Material

## A  Detailed Setup

The experiments in this study are conducted using consumer hardware, including a single NVIDIA RTX 3090 (GPU), an AMD Ryzen 9 5900X 12-Core Processor working at 3.70 GHz, and 32 GB of RAM. We use the PyTorch 2.0 framework. The environments used for the evaluation are partially adapted from the `ma-gym` library with the OpenAI Gym interface and PyGame as the rendering library.

## B  Hyperparameter Tuning

To ensure full reproducibility of the experiments in addition to releasing the code publicly, we describe here the procedure we employ for tuning the hyperparameters. The environment parameters used in the evaluation of our approach can be found in Table 1. We select ranges of optimality guided by the task requirements and explore the different combinations using Bayesian search methods. A comprehensive list of the tuned hyperparameters can be found in Table 2. For baseline execution, ATOC is configured with the same hyperparameters as those used in our solution for target network update and storage sizes, as ATOC is also an off-policy algorithm. With respect to IC3Net, an on-policy algorithm, we use the default hyperparameters without further tuning. The architectures of the networks of individual agents implementing the baseline algorithms are the same as in the corresponding original papers. On the other hand, IPPO agents were implemented using the same recurrent network architecture as our agents and suggested tuned hyperparameters for multi-agent setting. We tested both shared-weight and independent critic network implementations for it, without noticing a significant difference in terms of resulting performance.

Table 1: Environments parameters. (*) To enable a fair comparison between different experimental settings, we adopt the same map size with agents positioned strategically along four axes relative to the goal position. This prevents the sharing of useful information beyond predefined pairs of agents while allowing us to test our method at scale.

|  | Switch | Cooperative Load Transportation | | | Predator-Prey | | |
| --- | --- | --- | --- | --- | --- | --- | --- |
| Obs space type | Discrete | | Discrete | | | Discrete | | |
| Obs space size | 4 | | 30 | | | 77 | | |
| Action space type | Discrete | | Discrete | | | Discrete | | |
| Action space size | 5 | | 5 | | | 5 | | |
| Map size | 7x3 | | 16x10 (*) | | 12x12 | 14x14 | 16x16 |
| $N^o$ agents | 4 | 2 | 4 | 8 | 4 | 8 | 16 |
| $N^o$ other entities | - | 1 (load) | 2 (loads) | 4 (loads) | | 16 (prey) | |
| Step reward | 0 | | 0 | | | 0 | | |
| Intermediary reward | - | | 0.5 | | | 0.1 | | |
| Goal reward | 5 | | 5 | | | 5 | | |

Table 2: Network training hyperparameters.

|                            | Sw    | CLT     | PP    |
| -------------------------- | ----- | ------- | ----- |
| Optimizer                  |       | RMSprop |       |
| Q-net lr                   |       | 0.0001  |       |
| Coordinator lr             |       | 0.00005 |       |
| Weight decay               |       | 0.00001 |       |
| Beta1                      |       | 0.9     |       |
| Beta2                      |       | 0.99    |       |
| Gamma                      |       | 0.99    |       |
| Batch size                 |       | 512     |       |
| Recurrent steps            | 2     | 2       | 10    |
| Q update interval          |       | 50      |       |
| Coord update interval      |       | 50      |       |
| Target net update interval |       | 20000   |       |
| Min buffer size            | 1000  | 5000    | 5000  |
| Max buffer size            | 20000 | 20000   | 20000 |

## C   Architecture of the Neural Networks

Each agent network consists of a Feature Extractor module, a Q-network module, and Coordinator module:

- The Feature Extractor module comprises a MLP with two linear layers and ReLU activations. The layers map the input data to a latent representation of dimension 128.

- The Q-network module plays a key role in two phases of our architecture. In the first phase, we use a single layer GRU with hidden size 128 followed by a normalization layer and a linear layer to generate Q-values for the agent's action space based on the extracted features. In the second phase, we use a MLP network to map agent messages to an agent-specific representation space and another one to compute weights for the state-action values from the first phase. Each network consists of two 128-unit linear layers with ReLU, but with the second network having a Sigmoid activation function as the final activation.

- The Coordinator structure consists of a recurrent submodule that includes a BiGRU of size 128, which is responsible for processing concatenated messages. Next, a normalization layer is applied, followed by a two-layer MLP with ReLU nonlinearity. The forward and backward output at each step of the sequence is processed in this way, resulting in two logits used to generate the coordination mask for each agent.

## D   Additional Details about Data Processing

To compute the "smoothed" time series shown in the plot of Figure 3 relative to the Switch environment, we apply a rolling window of size 200 to smooth the mean reward data of 5 executions collected, and estimate the mean reward over time. The confidence interval bounds are computed by means of a rolling mean over the min and max values. For the results of the Cooperative Load Transportation and Predator-Prey environments, we consider the mean performance value at the end of the training with confidence intervals computed using an exponential moving average with smoothing value of 0.95.

## E   CoMIX Training Loop

The pseudocode of the CoMIX training loop is presented in Algorithm 1.

**Algorithm 1** CoMIX Training Loop

---

 1: Initialize weights for Q network and target Q network
 2: Initialize weights for Coordinator
 3: Initialize replay buffer
 4: **for** episode $\leftarrow 1$ to numEpisodes **do**
 5:     Initialize episode
 6:     **while** not done **do**
 7:         Receive observation
 8:         $Q_{self} \leftarrow Q_{policy\_1}(\text{observation})$
 9:         Receive messages from communication channel
10:         $c_i = Coord(\text{messages})$
11:         Optimization for Coordinator weights
12:         Filter the messages with coordination mask $c_i$
13:         $Q_{coord} \leftarrow Q_{policy\_2}(\text{filtered messages})$
14:         $Q_i \leftarrow Q_{self}Q_{coord}$
15:         Choose an action based on $Q_i$ values
16:         Execute the chosen action, and observe reward and if episode terminates
17:         Store transition data in replay buffer
18:         Sample a batch of transitions from replay buffer
19:         DDQN optimization for Q network weights
20:         **if** step count % target_update $= 0$ **then**
21:             Update target Q network weights with Q network weights
22:         **end if**
23:     **end while**
24: **end for**

---