# OpenReview forum: "CoMIX: A Multi-agent Reinforcement Learning Training Architecture for Efficient Decentralized Coordination and Independent Decision-Making"
_TMLR — Accepted by TMLR_

### Review · Reviewer_LiJe · 2024-03-15

**Summary Of Contributions:**

This paper proposes a new method called Coordinated QMIX (CoMIX) for cooperative MARL in the centralized training decentralized execution paradigm with a communication channel. The individual Q function in QMIX is modified to be a product of two functions, one that produces an initial Q value based on the agent's own state, while the other is a coordination/communication model that uses all of the initial Q values of all agents to produce a scalar coefficient that changes the final Q values of each agent. The method is evaluated on three gridworld environments with between 2 to 16 agents, and performs mostly favorably compared with independent PPO and two other MARL communication methods IC3Net and ATOC.

**Audience:**

Yes

**Claims And Evidence:**

Yes

**Requested Changes:**

0. Justify equation (7)
1. [Critical] Compare to QMIX
2. [Critical] Clarify how the Coordinator is trained (using loss (7) or loss (6) or combination of both)
    * [Critical] If Coordinator is trained by both (6) and (7), then the current ablation that removes the coordinator isn't enough to isolate the effect of loss (7). That means authors need to compare to an ablation of CoMIX where the coordinator is still present but the "self-supervision" loss equation (7) is removed, i.e. coordinator is trained only by the regular TD loss (6).
3. More analysis of the kinds of environments where one should or should not use CoMIX, compared to other methods.
4. [Critical] Some broad claims are not backed up by references, e.g.
    * intro paragraph 2 "In fact, coordination emerging from indiscriminate communication may not always be the best solution"
    * intro page 2 "For instance, in crowded or sparsely rewarded settings, enforcing top-down coordination on all agents may hinder convergence to an optimal joint policy."
    * intro page 2 "In contrast, allowing potential independent local choices, rather than mandating group coordination, can lead to emergent collaboration arising naturally where needed...", e.g. cite [1,2]
5. [Critical] Provide evidence for the claim "CoMIX’s ability to efficiently choose between collaboration and independent action", e.g. scenarios where the communication layer caused the CoMIX agents to modify their initial Q values to take more cooperative actions, and scenarios where the communication layer can be ignored and agents just use greedy individual actions, and also show how the baselines behave in these cases.

* [1] Inequity aversion improves cooperation in intertemporal social dilemmas. Hughes et al., NeurIPS 2018
* [2] Learning to Incentivize Other Learning Agents. Yang et al. NeurIPS 2020

**Strengths And Weaknesses:**

Strengths:

1. Extending QMIX by splitting the individual Q function into two parts and adding a multi-agent communication layer makes sense intuitively.
2. The investigation of communication disruption is novel and is a good step toward practicality.


Weaknesses:

0. The key question "Can agents acting in a shared environment with the ability to communicate autonomously learn coordinated behaviors..." is already answered in the positive by CTDE and many of the cited works on MARL communication, so it's not clear why the paper treats this as an open question.
1. The intuition behind the 2-step computation of the individual Q function is not backed up by more rigorous theory or analysis. In principle, having the communication layer at test time means CoMIX agents have more information than QMIX agents. But perhaps the mixer network in QMIX can achieve a similar effect during training, so that communication isn't needed at test time.
2. The writing in Section 4.1.2 is confusing. The Coordinator module is not yet defined at that point, so it is not clear how the $c_{i,j}$ are computed from the pairs of messages.
3. The "self-supervision" in equation (7) is insufficiently justified. Self supervision is usually understood as perturbing a sample and training a model to reconstruct the original sample based on context (e.g., BERT), thereby providing a context-based representation of individual parts of the sample (e.g. a word). Equation (7) is more like a contrastive loss that tries to increase the gap between the Q value with the actual message and a perturbed Q value from using another choice of message. The effect of this is not clear.
4. Not clear if the Coordinator parameters $\theta^C$ are trained purely by minimizing loss (7) or also by the regular TD error loss (6) or by some combination of both.
5. The method is an extension of QMIX, but experiments do not compare with QMIX.
6. Performance of CoMIX compared with the chosen baselines (IPPO, ATOC, IC3Net) are ambiguous. Sometimes a baseline is better, sometimes CoMIX is better. Maybe CoMIX is more suitable for some types of environments, but this isn't analyzed in more detail. So it's not clear when one would use CoMIX for a new environment.
7. Writing $Q_i = Q_{self}Q_{coord}$ is strange if we view $Q$ functions as having "physical units" of reward. Reward squared doesn't make sense, just as how writing $F_1 = F_2F_3$ where each $F_i$ is a force with units of Newtons doesn't make physical sense. The authors probably mean that $Q_{coord}$ is a dimensionless coefficient, not an actual action-value function.
8. Section 5.4.1 is speculation and doesn't add much to the paper.

---

> ### Author Response · Authors · 2024-04-07
> **Response to Reviewer (1)**
>
> We would like to thank the reviewer for their insightful comments. We have revised the paper accordingly - the new version has been uploaded in OpenReview. We would like to thank the reviewer for the comments and suggestions. We also incorporated small changes to the architecture of COMIX, which yields improved performance results.
>
> > The key question "Can agents acting in a shared environment with the ability to communicate autonomously learn coordinated behaviours..." is already answered in the positive by CTDE and many of the cited works on MARL communication, so it's not clear why the paper treats this as an open question.
>
> We agreed that the formulation of the research question was not particularly insightful and did not help highlight the novelty of our work. We have revised the formulation of the question following the suggestion of the reviewer and added new results and analysis to the manuscript. In particular, we have rephrased the research question to focus on the emergence of collaborative behaviour among agents acting in a shared environment. The new question is as follows:
> *“Can agents acting in a shared environment, through communication and without collaboration constraints, learn to balance coordinated and individual behaviors in order to successfully achieve a given goal?”*
>
> > The intuition behind the 2-step computation of the individual Q function is not backed up by more rigorous theory or analysis. In principle, having the communication layer at test time means CoMIX agents have more information than QMIX agents. But perhaps the mixer network in QMIX can achieve a similar effect during training, so that communication isn't needed at test time.
>
> We would like to thank the reviewer for this insightful observation. In general, CoMIX can be seen as an extension of QMIX with the addition of agent-to-agent communication capabilities. It has the same advantages and potential limitations in terms of applicability like QMIX in relation to the use of CTDE methods for training. Compared to other approaches, it is more robust in presence of potential communication issues and more efficient, since it selects only a subset of information on which the other agents have to act upon. Because of this flexibility, agents are able to choose either to act independently or to participate in collaboration with other agents.
>
> As suggested,  it is possible to assume that the Mixer network at training time can already enable the agent's network to learn relationships that make communication unnecessary at testing time. However, this would force agents to learn from “assumptions” made about other agents instead of from explicit communication, which is the main focus of the paper, and also would neglect the possibility of deploying the solution in the presence of agents not seen at training time.
>
> > The writing in Section 4.1.2 is confusing. The Coordinator module is not yet defined at that point, so it is not clear how the $c_{i,j}$ are computed from the pairs of messages
>
> We agree that the order in which we presented the design of Coordinator in Section 4.1.2 generates potential confusion in the reader. We have changed the text following the reviewer’s suggestion.
>
> > The "self-supervision" in equation (7) is insufficiently justified. [...] Equation (7) is more like a contrastive loss…
>
> In Section 4.2.2 we refer to the optimization of the Coordinator module as a process of self-supervision since we use the output of the model to optimize its component but as suggested this is not the canonical definition of self-supervision. We agree with the reviewer that this is confusing and changed the wording using “contrastive optimization” instead. Furthermore, we have changed the description to state more clearly how Eq.7 can provide a legitimate optimization signal for $\theta^C$, the parameters of the Coordinator. In particular we point out that the propagation of the error (the clipped advantage in $Q$-value) contributes to the probability of the message choice without affecting the $Q$-network weights directly.
>
> > Not clear if the Coordinator parameters are trained purely by minimizing loss (7) or also by the regular TD error loss (6) or by some combination of both.
>
> The optimization processes for the $Q$-network and the Coordinator are well distinguished and described in the respective sections. In Section 4.2.1 we describe the optimization of $\theta^Q$, which parametrizes only the Mixer and the individual $Q$-networks. The Coordinator, instead, is parametrized by $\theta^C$, optimized by means of Eq.7 as described in Section 4.2.2. We have clarified these points in Sections 4.2.1 and 4.2.2 accordingly.

---

> ### Author Response · Authors · 2024-04-07
> **Response to Reviewer (2)**
>
> > The method is an extension of QMIX, but experiments do not compare with QMIX.
>
> CoMIX extends QMIX by incorporating a communication step in the computation of the $Q$-value. As a result, when this communication step is removed, we revert back to the original implementation of QMIX. Although it was not explicitly mentioned in our original submission – and we have stressed this in our revision of the work –  the ablation of communication in the Experiments section, namely ‘CoMIX w/o comm’, corresponds to the QMIX implementation.
>
> > Performance of CoMIX compared with the chosen baselines (IPPO, ATOC, IC3Net) are ambiguous. [...] So it's not clear when one would use CoMIX for a new environment.
>
> Thanks to the comments of the reviewers, we have refined and expanded the evaluation section in our revised submission, emphasizing the strengths of the proposed method. Our approach is generally suitable for scenarios in which it is not viable to deploy a central entity to control individual behavior, but in which agents can effectively utilize a communication channel to coordinate their actions. The flexible and versatile implementation of our approach supports agents’ collaboration and independence at the same time. It is also robust against noisy communication channels that may hinder the performance of other methods. We have clarified this point in Section 5.4.3.
>
> > Writing $Q_i=Q_{self}Q_{coord}$ is strange
>
> We agree with the suggestion of the reviewer, since $Q_{coord}$, indeed, does not represent an action-value function but only calculates the weights to scale the state-action pairs. We have modified the notation to reflect its actual role.
>
> > Section 5.4.1 is speculation and doesn't add much to the paper.
>
> We would like to thank the reviewer for pointing out this issue. We agree that that section was not essential. We have removed the section from the current version of the manuscript
>
> **About Requested Changes:**
>
> We have addressed requested changes 0, 1, 2, 3 as commented above, and added missing references to back our claims as requested in 4.
>
> > Provide evidence for the claim "CoMIX’s ability to efficiently choose between collaboration and independent action"
>
> We provide an ablation analysis of the impact of the communication mechanism in Section 5.3 dedicated to analysis of the experimental results, showing quantitatively (Fig. 3) how its introduction positively affects the performance of CoMIX. In addition, we have expanded the section with a qualitative analysis, comparing the behavior of CoMIX agents in a scenario in which communication is possible and one in which it is not.

---

> > ### Comment · Reviewer_LiJe · 2024-04-14
> > **Acknowledgement of the author's reply**
> >
> > I appreciate the author's clear and detailed responses, clarification, and revision of the paper. My main concerns are addressed.

---

### Review · Reviewer_Km6W · 2024-03-25

**Summary Of Contributions:**

This work proposes Coordinated QMIX (CoMIX), a multi-agent reinforcement learning approach within the centralised training with decentralised execution paradigm that incorporates a communication step in the QMIX architecture and allows agents to trade-off collaborative and independent decision-making, when optimising their behaviour.

**Audience:**

Yes

**Claims And Evidence:**

No

**Requested Changes:**

Overall, I would advise the following:
- Solidify the targeted setting for CoMIX and make sure to investigate this in the experimental settings (e.g., settings requiring clear independent behaviour and only sparse collaborative moments)
- The proposed communication scheme is the novel point of the work, however, it is lost inside the QMIX architecture, with the mixer network. In order to highlight the novelty and impact of the communication channel and coordinator, can this module be used without the mixer, in a different architecture?

**Strengths And Weaknesses:**

Strengths:
- Addresses a challenging setting in MARL, namely when should agents coordinate in challenging settings, in potentially sparse interactions
- Proposes an interesting communication scheme based on observations and action intentions
- Fairly well written work

Weaknesses:
- Concerns on clarity and soundness

Conceptually, in comparison to QMIX, CoMIX introduces a communication step in the local agent Q networks (based on observation and next action sharing), together with a coordination module. This is claimed to allow agents to filter incoming information and tradeoff between taking independent or collaborative actions.
Major concern: In this context, what is the role of the Mixer network, and how does the end-to-end training procedure using the $Q^\text{TOT}$ signal fit in the CoMIX framework?

Section 4.2.2 was unclear. I failed to understand the motivation behind the inverted probabilities and the averaged pairwise communication losses. Also, how are the $n-1$ masks per agent impacting the training complexity?

I appreciate the ablation, efficiency and resilience studies. However, it is difficult to understand the explanation on the fine-tuning procedure (on the resilience study), as well as the procedure and results of Figure 4 (on the efficiency side).

Finally, I was wondering what is the impact on the exploration behaviour of the communication module in CoMIX, or the other way around (if agents commit to a certain action, but then take an exploratory step?).

- Significance concerns
For the baselines, it is correct to state that CoMIX without communication is QMIX?
Unless I am misunderstanding Figure 3, CoMIX does not seem to present performance improvements in the considered environments and does not scale well with the number of agents.


Other remarks:
- I did not find Fig. 1 b) to be clear, and if I understand correctly, it is a conceptual demonstration and not an actual slice from the inner workings of CoMIX?
- There are some vague statements that would use more support in the paper presentation:
    - "Incremental policy approach" - what is meant with this and how is this reflected in the technical constitution of CoMIX?
    - "balancing independence and collaboration" - this one a major motivation point for CoMIX, but it is not strongly reflected in the experiments, for example the environment selection.
    - the introduction and conclusion also discuss 'sparsely rewarded settings', but this is not investigated in the experiments
- There are also some minor errors present:
    - 'coordination among them become more...' -> becomes
    - 'it represents a fundamental aspects...' -> aspect
    - 'it represents...' and 'It is also critical' -> not clear what the 'it' refers to (perhaps coordination?)
    - 'more optimal solutions' -> I would consider 'optimal' as non-gradable, so perhaps 'better optimised' is more suitable
    - 'a solution that give more...' -> gives

---

> ### Author Response · Authors · 2024-04-07
> **Response to Reviewer (1)**
>
> We appreciate the reviewer's valuable feedback, including identifying typos in the text and we would like to thank him for the comments and suggestions. A revised version of the manuscript containing all the changes and new experimental results has been submitted in OpenReview. We also incorporated small changes to the architecture of CoMIX, which yields improved performance results.
>
> > [...] What is the role of the Mixer network, and how does the end-to-end training procedure using the $Q^{TOT}$ signal fit in the CoMIX framework?
>
> The observation made by the reviewer that “CoMIX without communication is QMIX” is indeed correct. We have clarified this point further in the revised version of the manuscript. We decided to adopt such a framework to be able to train a variable number of agents using a low computational footprint strategy, while still achieving a high level of performance. In fact, despite its limitations, QMIX has been shown to achieve state-of-the-art results on a variety of  MARL tasks, making it a suitable choice for our needs. We selected a state-of-the-art training framework since that is not the focus of this manuscript. The contribution of this work resides in the aspects related to communication. In a sense, the type of training framework and the presence (or not) of communication mechanisms can be considered orthogonal aspects; indeed, the Mixer network can be replaced with other training methods.
>
> More specifically, the Mixer network and thus the $Q^{TOT}$ signal are adopted in the CoMIX framework in order to enable end-to-end training of individual $Q$-networks. Given information from the environment and incoming filtered messages from other agents, the networks compute state-action $Q$-values as output and the $\theta^Q$ weights parameterizing the networks are optimized by Eq. 6. The Coordinator plays a passive role in this optimization process, since its $theta^C$ weights are optimized separately by means of Eq. 7.
> To clarify potential confusion concerning $\theta^Q$ and $\theta^C$ optimization,  we have modified Sections 4.2.1 and 4.2.2.
>
> > Section 4.2.2 was unclear. I failed to understand the motivation behind the inverted probabilities and the averaged pairwise communication losses. Also, how are the $n-1$ masks per agent impacting the training complexity?
>
> Since the Coordinator predicts probabilities of accepting or rejecting a message for each other agent, our method of optimization consists in considering the opposite choice $n-1$ times. The substantial computational impact is related to the calculation of the final state-action values for each different set of messages ($n-1$ times). Then, considering each maximum $Q$ as a signal of maximum future reward, we compare them with the $Q$ value obtained with unmodified probabilities predicted by the Coordinator. In the revised version of the manuscript we have simplified Section 4.2.2 in order to clarify this point. Specifically, instead of referring to probabilities we refer to a subset of messages – direct consequence of probability prediction – which more directly influence the computation of the $Q$-value.
>
> > [...] It is difficult to understand the explanation on the fine-tuning procedure (on the resilience study), as well as the procedure and results of Figure 4 (on the efficiency side).
>
> We have revised Section 5.4.2, aiming to provide a clearer explanation of the fine-tuning procedure.
>
> > What is the impact on the exploration behaviour of the communication module in CoMIX, or the other way around (if agents commit to a certain action, but then take an exploratory step?)
>
> We would like to thank the reviewer for this observation, since it touches upon a very interesting aspect of the design of COMIX, namely the mutual interactions of the different mechanisms at the basis of our approach. The exploration behavior of the communication module does have an impact, which is dependent on the tasks and environments. The communication mechanism relies on exploration, but it might affect the performance of the algorithm if, for example, the exploration probability of the communication mechanism is too high, as for other RL algorithms. However,  our results show that the introduction of the communication mechanism is beneficial. In order to fully address the concern of the reviewer  we have added a paragraph about a focused comparison between the behavior of "CoMIX" and "CoMIX w/o comm" (i.e., COMIX without communication) in Section 5.3.

---

> ### Author Response · Authors · 2024-04-07
> **Response to Reviewer (2)**
>
> > I did not find Fig. 1 b) to be clear, and if I understand correctly, it is a conceptual demonstration and not an actual slice from the inner workings of CoMIX?
>
> The intent of Fig. 1 b is to graphically express the idea of how communication affects the evaluation of possible actions in a certain state. Should the message conveyed be still confusing, we are happy to remove it altogether. In order to improve the figure, in the revised version of the manuscript, we have corrected and updated the notation. We have also corrected a graphical error that we noticed in the previous version of the figure.
>
> > "Incremental policy approach" - what is meant with this and how is this reflected in the technical constitution of CoMIX?
>
> The term was referring to the two incremental steps of calculating final state-action values. We agree with the reviewer that "incremental policy approach" is undoubtedly strange as a term. We have changed the term to "incremental approach."
>
> > "balancing independence and collaboration" - this one a major motivation point for CoMIX, but it is not strongly reflected in the experiments, for example the environment selection.
>
> The meaning of the phrase refers to the method's ability to act flexibly both in independent situations and by adopting collaborative behaviors. We highlight the emergence of such situations in the description of environments and demonstrate the ability to deal with them successfully by providing quantitative and qualitative results in Section 5.3. In addition, we have included a paragraph in Section 5.3 to analyze agents' behaviors of "CoMIX" versus "CoMIX w/o comm."
>
> > the introduction and conclusion also discuss 'sparsely rewarded settings', but this is not investigated in the experiments
>
> By "sparsely rewarded settings" we refer to the "Switch" environment, in which agents are rewarded only if they successfully complete the task, and to "Predator-Prey," in which major rewards are given only upon completion of the difficult task of surrounding a prey. We have clarified these points in Sections 5.1.1 and 5.1.3.
>
> **About Requested Changes:**
>
> We have included details related to environment selection on the base of methods characteristics in Section 5.4.3 and discussed above in the comments the choice of adopting QMIX as a training framework.

---

> > ### Comment · Reviewer_Km6W · 2024-04-19
> > **Minor further remarks**
> >
> > I thank the authors for all the clarifications. I have read the responses, as well as the revised manuscript, and had a few questions/remarks left.
> >
> > - I could not identify the changes to the architecture of CoMIX. Could you provide more details on that update?
> >
> > - I think further clarification for how each $w_i$ (Eq. 7) is obtained would be helpful. I find this sentence insufficient:
> > "The additional weighting term wi is obtained by means of QMIX, which, by design, maps states into a set of values in the hidden space for each individual agent."
> >
> > - Table 1, clarify what the values represent, since it is not uniform over all the environments.
> >
> > - Section 5.4.3 not supported by any empirical or theoretical evidence, in its current form. I think grounding the discussion using the experimental results to support the claims would be beneficial.

---

> > > ### Author Response · Authors · 2024-04-20
> > > **Response to Reviewer**
> > >
> > > We thank the reviewer for their remaining comments. Following the additional remarks we submitted an updated version of the manuscript, while our responses to the specific points are as follows:
> > >
> > > >  I could not identify the changes to the architecture of CoMIX. Could you provide more details on that update?
> > >
> > > In addition to the modifications to the training hyperparameters, which are detailed in the Supplementary Material, the relevant description can be found in Section 4.2.3.  We have realized that unfortunately we did not highlight that section in yellow as well - that was our mistake, we apologize for this. The major changes are the following: the use of normalization layers and the substitution of Gumbel-Softmax as activation function on the Coordinator output.
> > >
> > > > I think further clarification for how each $w_i$  (Eq. 7) is obtained would be helpful.
> > >
> > > Given the joint state of all agents, by design, the first layer of the Mixer network computes a projection matrix to map each individually estimated $Q$-value in a shared hidden space. The second layer uses the same approach  to reduce the dimensionality of the intermediary value to a single value $Q^{TOT}$. We leverage the monotonicity constraints of the Mixer to compute $w_i$ as the per-agent averaged projection weights, which scale their estimated $Q$-values based on the current joint state. We have extended the corresponding paragraph in the manuscript  with a more complete description of the derivation of $w_i$. The text in the paper has been changed accordingly. The changes are in light blue.
> > >
> > > > Table 1, clarify what the values represent, since it is not uniform over all the environments.
> > >
> > > We would like to thank the reviewer for this comment. We agree that the table was not sufficiently clear. We have revised the caption of Figure 3 to provide a clearer explanation of what the reported metric signifies for each environment. Additionally, we reported the definitions of the performance metrics in the table caption. The changes are in light blue.
> > >
> > > > Section 5.4.3 not supported by any empirical or theoretical evidence, in its current form. I think grounding the discussion using the experimental results to support the claims would be beneficial.
> > >
> > > Following the reviewer's suggestion, we have expanded Section 5.4.3 by adding a discussion of (and references to) experimental results, in order to ground our discussion of CoMIX characteristics. We agree that this definitely helps in strengthening this section of the paper. The changes are in light blue.

---

> > > > ### Comment · Reviewer_Km6W · 2024-04-21
> > > > **Thank you**
> > > >
> > > > I thank the authors for addressing the additional questions and concerns, I have no further remarks.

---

### Review · Reviewer_qAXm · 2024-03-29

**Summary Of Contributions:**

The authors are interested in cooperative multi-agent RL, where agents see states and take actions independently but have a single shared reward. A common paradigm in this type of setting is “centralized training, decentralized execution”. A popular CTDE approach is known as QMIX, where agents’ individual Q-functions (trained according to the standard RL loss) are aggregated in a particular way into a combined Q-function which is used during training (at test time, policies maximizing the individual Q-functions are used). To this approach, the authors add a “coordinator” module, which can accept messages from agents and selectively send them to other agents to improve performance (the coordinator observes all messages to be sent and masks out some of them). The coordinator network is trained using a self-supervised “contrastive” loss to maximize the degree to which its mask helps increase Q-values. The authors show in experiments that their approach can perform competitively, they do some ablation studies on the coordinator, study how efficient the communications are, and consider a case where messages can be disrupted (this last experiment also shows that the message sharing is actually being used).

**Audience:**

Yes

**Broader Impact Concerns:**

No broader impact concerns.

**Claims And Evidence:**

Yes

**Requested Changes:**

I think it would be helpful for the authors to go through the paper again and look for places where they have been vague or left out technical details -- my guess is these can be corrected very easily.

**Strengths And Weaknesses:**

Strengths:

- The idea makes intuitive sense, and the authors establish that it does seem to be “actually working” — the agents are making use of the communication, and the coordinator is actually doing something to make sure useful messages get sent.

Weaknesses:

- Many of the ideas and experiments are verbally described but don’t include key details (i also checked the appendix). One example: the authors just refer to “a small MLP” in the appendix without specifying the size.

Both strengths and weaknesses:
- The experimental results are mixed. The authors are quite clear and honest about this. It is difficult to compare RL algorithms with very different properties, and just because this approach does not conclusively win out over all competitors, does not mean that there is no contribution here.

---

> ### Author Response · Authors · 2024-04-07
> **Response to Reviewer**
>
> We would like to thank the reviewer for their comments and suggestions. A revised version of the manuscript containing all the changes and new experimental results has been submitted in OpenReview. We also incorporated small changes to the architecture of COMIX, which yields improved performance results.
>
> > Many of the ideas and experiments are verbally described but don’t include key details (i also checked the appendix). One example: the authors just refer to “a small MLP” in the appendix without specifying the size.
>
> In an attempt to address the reviewer's comment, in our revision we have clarified the steps about the fine-tuning procedure in Section 5.4.2; we have updated and included missing/additional details  about training in Section 4.2.3; and we have added a more comprehensive description of the COMIX architecture in Section C of the Supplementary Material.

---

### Decision · Action_Editor_4rKz · 2024-05-12

**Recommendation:** Accept as is

**Comment:**

All reviewers are supportive about the method and the experimental findings.

**Audience:**

The paper would be of interest to the multi-agent RL community.

**Claims And Evidence:**

This paper proposes a new method called COMIX for multi-agent reinforcement learning (MARL). The method extends QMIX by incorporating a additional communication channel between agents, which gives a multiplicative term on each agent's own Q function. Experimentally, the approach is shown to achieve improved performance over existing methods on several collaborative MARL tasks.